# Modelling the impact of interventions on imported, introduced and indigenous malaria infections in Zanzibar, Tanzania

Aatreyee M. Das [1,2] ✉, Manuel W. Hetzel [1,2], Joshua O. Yukich [3], Logan Stuck [3,6,7], Bakar S. Fakih[1,2,4], Abdul-wahid H. Al-mafazy[5,8], Abdullah Ali[5] & Nakul Chitnis [1,2] ✉

Malaria cases can be classified as imported, introduced or indigenous cases. The World Health Organization's definition of malaria elimination requires an area to demonstrate that no new indigenous cases have occurred in the last three years. Here, we present a stochastic metapopulation model of malaria transmission that distinguishes between imported, introduced and indigenous cases, and can be used to test the impact of new interventions in a setting with low transmission and ongoing case importation. We use human movement and malaria prevalence data from Zanzibar, Tanzania, to parameterise the model. We test increasing the coverage of interventions such as reactive case detection; implementing new interventions including reactive drug administration and treatment of infected travellers; and consider the potential impact of a reduction in transmission on Zanzibar and mainland Tanzania. We find that the majority of new cases on both major islands of Zanzibar are indigenous cases, despite high case importation rates. Combinations of interventions that increase the number of infections treated through reactive case detection or reactive drug administration can lead to substantial decreases in malaria incidence, but for elimination within the next 40 years, transmission reduction in both Zanzibar and mainland Tanzania is necessary.

Globally, the case incidence of malaria has fallen from around 81 cases per 1000 population at risk from the year 2000 to 59 in the year 2020. Within the same time frame, deaths per 100,000 population at risk have halved, falling from 30 to 15[1]. As the burden of the disease falls, the number of countries looking to eliminate malaria grows. The World Health Organization (WHO) defines malaria elimination as the interruption of local transmission of a specified malaria parasite species in a defined geographical area as a result of deliberate activities[2]. WHO defines the interruption of local transmission as the reduction to zero incidence of indigenous cases, where it classifies *Plasmodium falciparum* malaria cases into the following categories: imported, introduced, indigenous, and induced, as defined in Table 1. Certification of malaria-free status by WHO requires the country to show three years of zero indigenous cases[3].

So far, WHO has certified 40 countries as having eliminated malaria, with another 61 classified as either a country where malaria

[1]Swiss Tropical and Public Health Institute, Allschwil, Switzerland. [2]University of Basel, Basel, Switzerland. [3]Center for Applied Malaria Research and Evaluation, Department of Tropical Medicine, Tulane University School of Public Health and Tropical Medicine, New Orleans, LA, USA. [4]Ifakara Health Institute, Dar es Salaam, United Republic of Tanzania. [5]Zanzibar Malaria Elimination Programme, Zanzibar, United Republic of Tanzania. [6]Present address: Amsterdam Institute for Global Health and Development Amsterdam, Amsterdam, Netherlands. [7]Present address: Amsterdam University Medical Centers, Amsterdam, Netherlands. [8]Present address: Office of the Chief Government Statistician (OCGS), Zanzibar, United Republic of Tanzania. ✉e-mail: aatreyee.das@swisstph.ch; nakul.chitnis@swisstph.ch

**Table 1 | WHO classification of malaria cases[3]**

| Term | Description |
|------|-------------|
| Imported case | Malaria case or infection in which the infection was acquired outside the area in which it is diagnosed |
| Introduced case | A case contracted locally, with strong epidemiological evidence linking it directly to a known imported case (first-generation local transmission) |
| Indigenous case | A case contracted locally with no evidence of importation and no direct link to transmission from an imported case |
| Induced case | A case the origin of which can be traced to a blood transfusion or other form of parenteral inoculation of the parasite but not to transmission by a natural mosquito-borne inoculation |

Furthermore, WHO defines a case as the occurrence of malaria infection in a person in whom the presence of malaria parasites in the blood has been confirmed by a diagnostic test[3]; therefore cases are defined on infection status and not on clinical symptoms.

never existed or where malaria disappeared without specific measures[4]. In 93 countries, malaria remains endemic, though 47 of these countries reported fewer than 10,000 cases in 2020[1]. As countries and regions head towards elimination, the focus of malaria programmes typically shifts from reducing the burden of the disease to reducing the rate of malaria transmission, finding and treating each remaining infection, and preventing the re-establishment of local transmission.

Since interventions do not have the same effect on the different categories of cases, different intervention approaches may be required depending on the composition of cases in a particular setting. Previous models of malaria importation have examined the presence of sources and sinks of malaria within a country[5,6], the proportion of detected infections that must be imported infections to ensure that each infection typically leads to fewer than one subsequent infection[7], and the reproduction number in the absence of importation[8]. Wesolowski et al. (2012) and Ruktanonchai et al. (2016) studied the movement of malaria infections within Kenya and Namibia using mobile phone usage data to infer where malaria would not be sustained without ongoing importation of infections. Churcher et al. (2014) used branching process theory to model the total number of infections stemming from a single malaria infection and used this to show that the reproductive number is likely to be below 1 in Eswatini. Le Menach et al. (2011) used a combination of mobile phone data and ferry traffic data to estimate the per capita malaria importation rate for Zanzibar, Tanzania. From this, they concluded that the reproduction number for malaria was below 1 on both major islands of Zanzibar and that typically around 1.6 cases were imported from mainland Tanzania per 1000 inhabitants per year. However, they assumed a constant importation rate and only importation from mainland Tanzania, excluding the movement of infections between the islands.

Zanzibar is a semi-autonomous archipelago of islands in the Indian Ocean just south of the Equator. It consists of two main islands, Unguja and Pemba. Unguja has a population of close to a million, is more urban and has stronger connections and more movement with mainland Tanzania. Pemba has less than half the population of Unguja, is conversely more rural and has fewer connections with the mainland.

Zanzibar has seen a decline in malaria transmission since the year 2000 due to the intensive use of vector control and passive surveillance efforts[9]. However, progress has stagnated since around 2007, with malaria persisting at a low prevalence on both main islands. Reactive case detection (RCD), the active search for malaria infections following the detection of a clinical index case at a health facility, was introduced in 2012 to help find malaria infections within the community, particularly those that may be asymptomatic and thus missed by passive surveillance. In Zanzibar, ~35% of index cases are followed up at their household (referred to as the index household) within 3 days. Within the index household, everyone who consents is tested with a rapid diagnostic test (RDT) and those found to be positive for malaria are treated. This RDT was estimated to have a sensitivity of 34% as compared to quantitative polymerase chain reaction (qPCR). Previous modelling studies have highlighted that improvements to RCD and sustaining current levels of vector control and passive surveillance are likely insufficient for achieving elimination[10], and imported infections need to be targeted to prevent chains of transmission[8,10]. However, all

these studies defined elimination as zero malaria infections, irrespective of their classification, which is not realistic in areas with regular movement of people to and from neighbouring regions with ongoing endemic transmission. To our knowledge, no prior studies have modelled imported, introduced, and indigenous infections explicitly and examined the impact of interventions on these three categories of infections; therefore no previous work has been able to model the probability of elimination as defined by WHO.

In this study, we explicitly model imported, introduced and indigenous cases separately to model the feasibility of achieving three years with no indigenous cases with current and potential future interventions to achieve the WHO standard for malaria-free certification. We do not include induced cases because they are responsible for less than 0.1% of all classified cases in Zanzibar (Abdul-wahid Almafazy, personal communication). We parameterise the model with data from 2017–18 from Zanzibar and analyse it to infer an estimate for the proportions of each category of infections on Pemba and Unguja. We then use this model to examine the impact of combinations of interventions such as improvements to reactive case detection (RCD), increasing the number of clinical cases detected in health facilities, switching to reactive drug administration (RDA), and treatment of imported infections. We also consider the impact of further reductions in transmission rates, both on Zanzibar and on the mainland, although we did not explicitly model the interventions that would lead to the reductions. The structure of this model allows us to explicitly model the probability of achieving the WHO definition of elimination—three years with zero new indigenous infections—as well as investigating the resulting changes in incidence on Zanzibar.

We follow WHO terminology in defining a malaria case as anyone infected with *P. falciparum* parasites, including both symptomatic and asymptomatic infections. However, we assume that diagnosis of cases only occurs in the patch of residence so we classify cases relative to their patch of residence: therefore we define imported infections as infections acquired when away from the area of residence; introduced infections as infections stemming from an imported infection, or from an infected visitor visiting the area of residence of the introduced infection; and indigenous infections as infections stemming from introduced or other indigenous infections. Thus, our definition of imported cases differs slightly from the WHO definition, as infected visitors are not counted as imported cases in the model (they would be classified as either an imported, introduced or indigenous case in their area of residence depending on where they acquired the infection). Our definition of introduced and indigenous cases match the definitions used by WHO, although in our simulations we have knowledge of the position of cases in the chain of transmission, which is not always known by elimination programmes when classifying cases.

## Results

Using our model, we estimate that 88% of new infections on Pemba are indigenous infections, 8% are introduced infections, and 4% are imported infections (Fig. 1). On Unguja, we estimate that 56% of new malaria infections are indigenous infections, 25% are introduced infections, and 18% are imported infections. These results are not directly estimated from local case notification data, but rather an

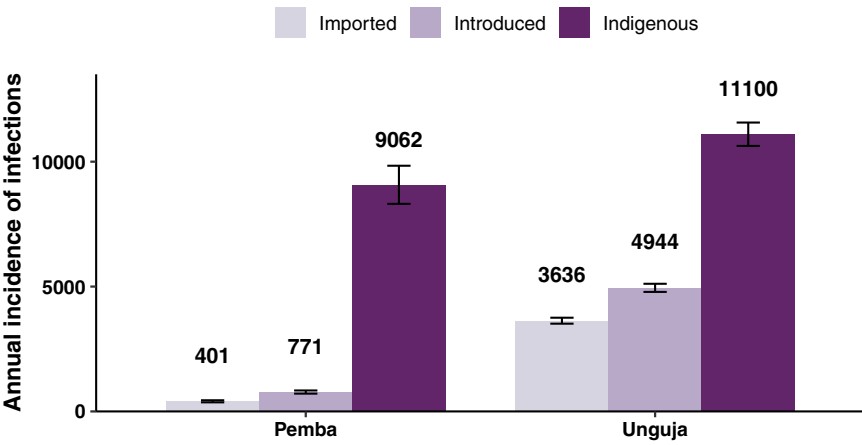

**Fig. 1 | Median annual incidence of imported, introduced and indigenous malaria cases at baseline.** Median annual incidence of imported, introduced, and indigenous infections on Pemba and Unguja at baseline. The height of the bar represents the median value across $n = 500$ simulations. The error bars represent the 95% prediction interval of the annual incidence.

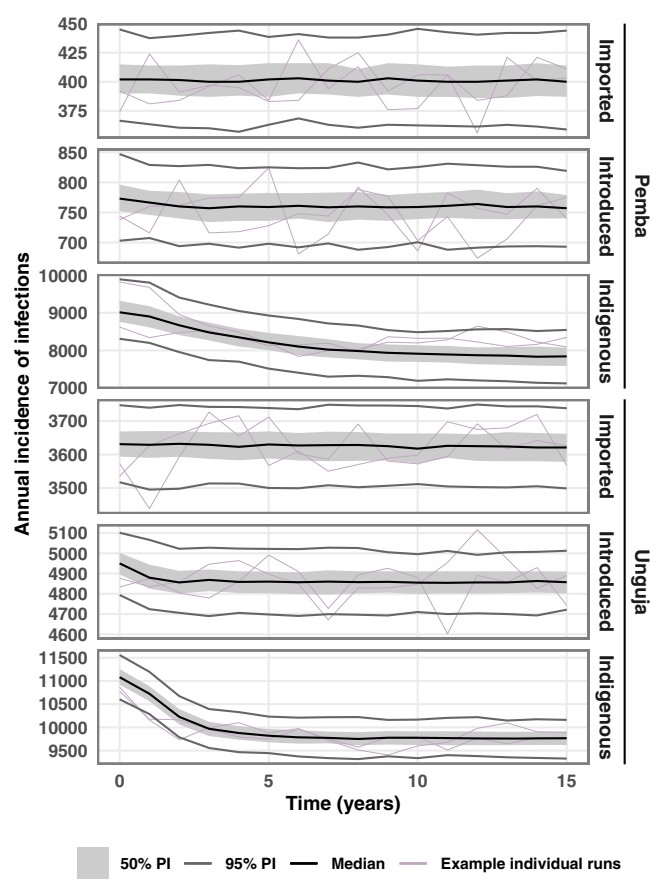

**Fig. 2 | Simulated annual malaria incidence after switching from reactive case detection (RCD) to reactive drug administration (RDA).** Time-series plot showing the median annual incidence of malaria infections across $n = 500$ stochastic simulations, after switching from RCD to RDA in year 0. Grey shaded area indicates the interquartile prediction interval, and the grey lines indicate the 95% prediction interval of simulation results. Purple lines are examples of individual runs. 'PI' stands for prediction interval.

output of the model, arising from the travel history of survey respondents and the prevalence of malaria in the areas visited.

Previously, a simpler version of this model found the calibrated values for the effective daily transmission rate for each infected individual ($\beta$) were 0.0048 day$^{-1}$ (95% CI: 0.0044–0.0050) on Pemba and 0.0037 day$^{-1}$ (95% CI: 0.0025–0.0.047) on Unguja[10]. The controlled reproductive number, given by the transmission rate divided by the recovery rate, was estimated to be 0.95 (95% CI: 0.88–1.00) on Pemba and 0.74 (95% CI: 0.50–0.94) on Unguja. These results remain unchanged by the extension of the model.

Removing RCD entirely is expected to lead to an increase in incidence of 10% in Pemba and 5% in Unguja. Switching from RCD to RDA is expected to lead to the treatment of approximately three times as many infections in the population for a given malaria prevalence, since RDTs currently miss approximately two-thirds of qPCR-detectable infections[11]. In the model, we observe 12% fewer new infections in Pemba and 7% fewer new infections in Unguja when we switch from RCD to RDA. In Fig. 2, we show time-series plots for the impact of switching from RCD to RDA at year 0 on the three categories of infections, since this is an intervention that is currently being considered for implementation by the Zanzibar Malaria Elimination Program (ZAMEP). The impact of switching to RDA on the incidence of imported cases is minimal, as transmission for these cases typically occurs on mainland Tanzania, and RDA is being implemented in Zanzibar. The impact on the incidence of introduced cases is small, and the impact on indigenous cases is substantial on both Pemba and Unguja, as these transmission events occur on Zanzibar and so are reduced by the shift from RCD to RDA.

Figure 3 shows the incidence of indigenous infections per 10,000 population in the 15th year from the implementation of interventions. Most RCD-related interventions have a similar impact on the incidence of indigenous infections (Fig. 3a). Across all three categories of infections, increasing follow-up of index cases from 35% to 100% is estimated to lead to an incidence reduction of 12% in Pemba and 7% in Unguja. Similarly, the median drop in incidence from a three-fold increase in the treatment-seeking rate is estimated to be 12% in Pemba and 8% in Unguja. Including 100 neighbours in RCD is expected to have a smaller impact than other RCD-related interventions, with a 6% reduction in incidence in Pemba and a 4% reduction in Unguja. Treating infected travellers has the largest impact on transmission, with a 90% treatment proportion leading to an 85% reduction in incidence on Pemba, and an 89% reduction on Unguja.

Combining interventions can have a multiplicative effect on the reduction in incidence. Figure 3b shows the impact of adding in new interventions on top of existing ones. Even without treating travellers, a 59% reduction in incidence amongst Pemba residents and a 40% reduction amongst Unguja residents can be achieved through the use of RDA with 100% follow-up of index cases, including 100 neighbours

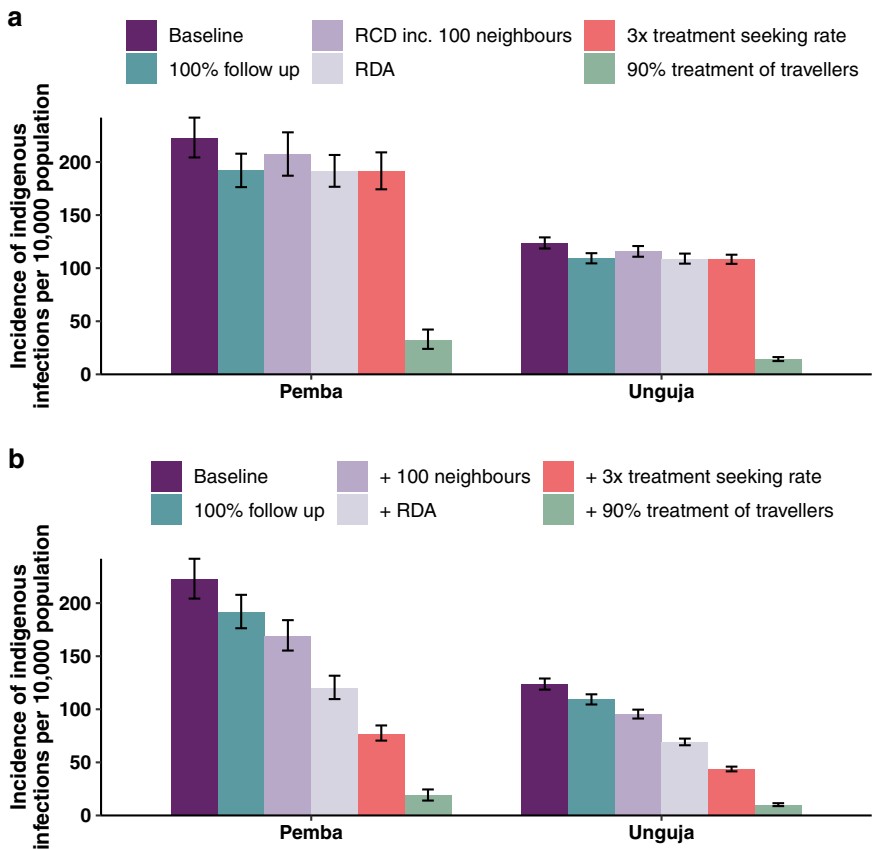

**Fig. 3 | Simulated annual malaria incidence 15 years after the start of interventions.** Median annual incidence of indigenous infections per 10,000 population in the 15th year after the start of interventions. The height of the bar represents the median value across $n = 500$ simulations per intervention scenario. The error bars represent the 95% prediction interval in the annual incidence. 'Baseline' refers to RCD with 35% follow-up of index cases at the index household. **a** Bar plot showing the final incidence after the implementation of each intervention on its own ($n = 500$ simulations per scenario). **b** Bar plot showing the final incidence when, going from left to right, each new intervention is layered on top of the last intervention ($n = 500$ simulations per scenario). 'RCD' stands for reactive case detection. 'RDA' stands for reactive drug administration.

in RDA, and increasing the treatment-seeking rate so that three times as many index cases are typically found in health facilities for a given malaria prevalence. Time-series plots of incidence for individual and combinations of interventions can be found in Figs. S8–S12 in the Supplementary Information.

We then considered the impact of further reducing the transmission rate on Pemba and Unguja. We find a 50% reduction in the transmission rate is expected to lead to a 89% drop in incidence on Pemba and a 62% drop in incidence on Unguja. We additionally investigate the likelihood of reaching zero indigenous infections over three consecutive years. Figure 4 shows the percentage of the 500 simulations that reach zero indigenous infections over three years at each time point, defined as reaching elimination. When there is no transmission reduction, even when 100% of infected travellers are treated, by the 40th year, 24% of simulations reached elimination in Unguja and 1% of simulations reached elimination in Pemba. Even with large reductions in transmission on Zanzibar, we see high probabilities of elimination only when all infected travellers are treated. This is due to the large numbers of imported infections and introduced infections stemming from visitors to both islands, but especially Unguja. Thus, even when 90% of travellers are treated, there are still sufficient numbers of imported infections that lead to onward transmission and eventually a handful of indigenous infections per year. Again, the results of combining all previously mentioned interventions with treatment of travellers and reductions in transmission can be found in Fig. S13 in the Supplementary Information. We find combining treating 90% of travellers with a 90% reduction in transmission leads to a 99.5%

reduction in incidence on Pemba and a 97.9% reduction in incidence on Unguja. The controlled reproduction number is below 1 for both islands, and this suggests that elimination should be achieved in the absence of importation. This is observed when the model is run for a longer period of time than 40 years (Fig. S15 in the Supplementary Information). Within 100 years of treating all infected travellers from mainland Tanzania, both islands reach almost 100% probability of reaching elimination.

As giving treatment or chemoprophylaxis to 100% of travellers is difficult to achieve, we also considered a potential reduction in malaria transmission on mainland Tanzania, thus reducing the number of imported infections arriving on Zanzibar. As shown in Fig. 5, a combination of a reduction in transmission on mainland Tanzania and on Zanzibar could lead to elimination on both Pemba and Unguja. The results from combining all previous interventions with transmission reduction on Zanzibar and mainland Tanzania can be found in Fig. S14 of the Supplementary Information. The probability of elimination after 40 years is estimated to be 31% on Pemba and 71% on Unguja when there is a 30% reduction in transmission on mainland Tanzania but no reduction in transmission on Zanzibar, and all RCD-related interventions are set to the maximum value given in Table 2.

In addition, the impact of changing the intervention parameters one at a time to see the impact on the final incidence of indigenous infections is explored in section S2.1 of the Supplementary Information. Increases in RCD-related interventions were found to lead to a linear decrease in malaria incidence, while the relationship between transmission reduction on Zanzibar and malaria incidence was found

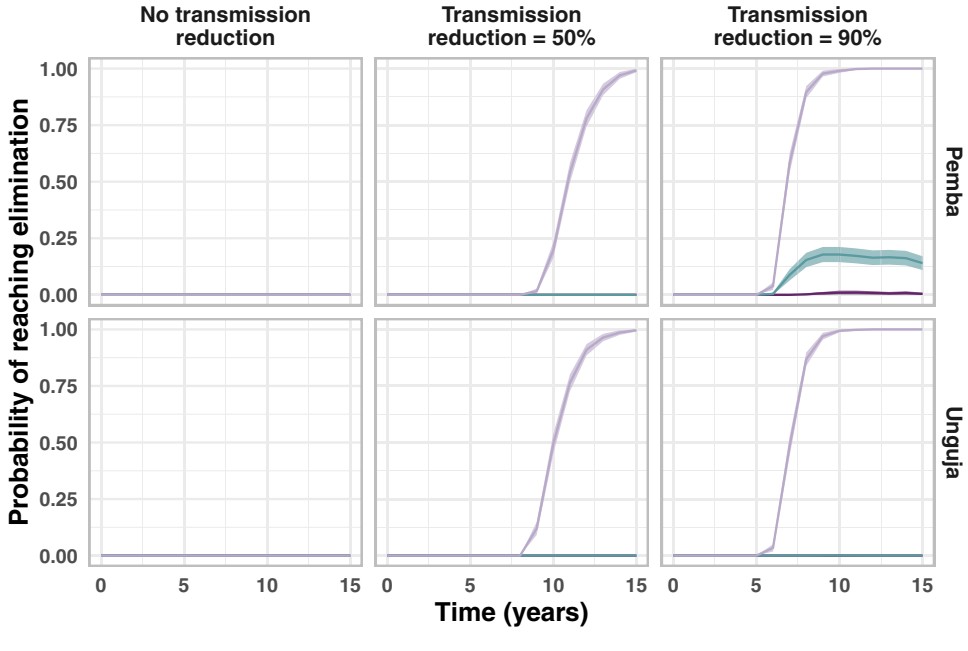

**Fig. 4 | Proportion of stochastic simulations reaching elimination upon treating infected travellers and reducing transmission rates across Zanzibar.** The central line is the proportion of $n = 500$ simulations that reached elimination (3 years with zero indigenous infections). The shaded area indicates 95% confidence interval (calculated assuming a binomial proportion using a Normal approximation interval). We assume that only the baseline interventions (RCD for 35% of cases arriving at a health facility at the index household level only) are present and then simulate reducing the malaria transmission rate on Zanzibar and treating a proportion of infections imported from mainland Tanzania.

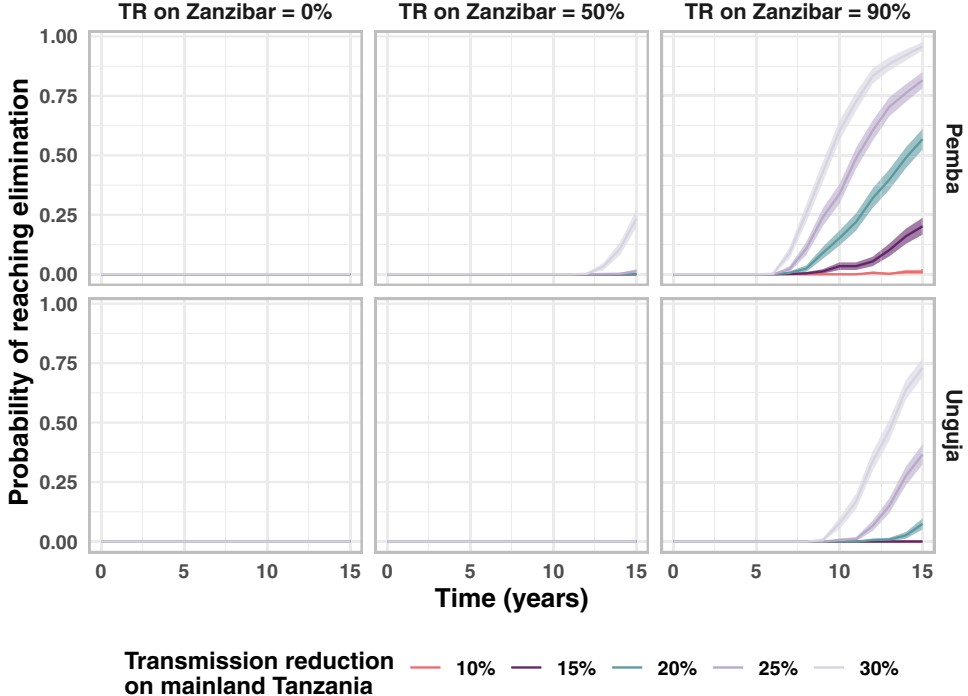

**Fig. 5 | Proportion of stochastic simulations reaching elimination when transmission reductions on Zanzibar and mainland Tanzania are combined.** The central line is the proportion of $n = 500$ simulations that reached elimination (3 years with zero indigenous infections). The shaded area indicates 95% confidence interval (calculated assuming a binomial proportion using a Normal approximation interval). These simulations consider that baseline interventions are in place (RCD with follow-up of 35% of cases at the index household level) and consider the impact of reducing the transmission rate on mainland Tanzania and combining this with a reduction in the transmission rate (TR) on the islands of Zanzibar. Interventions are introduced at time point 0.

**Table 2 | Baseline and intervention values for interventions simulated**

| Intervention | Baseline value | Intervention values |
|---|---|---|
| RCD follow-up | $\eta = 35.3\%$ | $\eta = [0\%, 100\%]$ |
| Increase in treatment-seeking rate | No increase | 100% increase, 200% increase |
| RCD including follow-up of $v$ neighbours | $v = 0$ neighbours | $v = [20$ neighbours, 100 neighbours$]$ |
| Switching from RCD to RDA (modelled as change in test sensitivity, $\rho$) | $\rho = 34\%$ | $\rho = 100\%$ |
| Treating a proportion of infections brought on to Zanzibar | Prop. treated = 0 | Prop. treated = [0.25, 0.50, 0.75, 0.90, 1] |
| Reductions in the malaria transmission rate on Zanzibar | $r_{Zanzibar} = 0$ | $r_{Zanzibar} = [0.25, 0.50, 0.75, 0.90, 1]$ |
| Reductions in the malaria transmission rate on mainland Tanzania | $r_{Mainland} = 0$ | $r_{Mainland} = [0.10, 0.15, 0.20, 0.25, 0.30]$ |

'Zanzibar' refers to both Pemba and Unguja.

to be highly non-linear, with a small reduction in the transmission rate leading to large decreases in malaria incidence.

## Discussion

We developed a model for estimating the proportions of infections observed in a region that are imported, introduced and indigenous, based on malaria prevalence and human movement data. This model can be applied to different settings and adapted to suit local interventions in place. We used this model to examine the role of imported infections in Zanzibar, a low-prevalence region with substantial importation of malaria infections and a well-established RCD programme.

The malaria situation is quite different between the two major islands of Zanzibar, and so the intervention effects also differ across the two islands. In Unguja, and to a lesser extent on Pemba, repeated importation of infections and local transmission from infected visitors is driving malaria persistence. Improvements in RCD, coupled with treatment of travellers, could lead to substantial reductions in the incidence of malaria infections, including indigenous infections. In addition to this, RCD is a useful surveillance tool that can be used for confirming the lack of indigenous infections and allowing for certification of malaria elimination. However, the large number of imported and introduced infections estimated in the model means that unless all infections coming from mainland Tanzania to Zanzibar can be treated or prevented, elimination is unlikely to be reached in Zanzibar, even though very low incidence levels can be reached. Instead, our results suggest that the pursuit of malaria elimination must be a coordinated effort on a national scale. Simulated decreases in the transmission rates on both Zanzibar and mainland Tanzania led to the largest reduction in malaria incidence and the highest likelihood of achieving malaria elimination on Zanzibar. Given that insecticide-treated nets and indoor residual spraying are already widely deployed in Zanzibar, further decreases in transmission rates may be difficult, but could potentially be achieved through novel supplementary vector control interventions such as volatile pyrethroid spatial repellents, odour-baited traps, and attractive targeted sugar baits. Transmission reduction could also be achieved with reactive vector control, which has shown promise in a field study in Namibia, especially when used in combination with RDA, and could be considered for deployment in a setting like Zanzibar[12].

Furthermore, these results assume all passive surveillance and vector control measures that are already in place are maintained, and that there is no significant malaria importation from outside of mainland Tanzania. Given that elimination is not currently certified by WHO at sub-national level, Zanzibar could only become certified by WHO when mainland Tanzania also has no community transmission and an application for elimination certification could be made for the entire United Republic of Tanzania[13].

Within the results, we observe that while RCD-related parameter values are higher in Unguja (e.g. a higher treatment-seeking rate, larger targeting ratio), and so the total rate of removal of infections ($\varphi$) is higher on Unguja, removing RCD (modelled as a counterfactual scenario) would lead to a larger relative increase in malaria incidence on

Pemba than on Unguja. We expect this is due to the higher transmission rate on Pemba than on Unguja. This highlights that even if RCD does not necessarily find and remove many cases, the effects of RCD compound over time and it can still have a substantial effect, particularly in higher transmission settings. However, the larger proportion of imported infections and smaller proportion of indigenous infections on Unguja suggests importation plays a larger role in sustaining transmission on Unguja than on Pemba. Treating infections in travellers is expected to have a large effect on malaria incidence, but there may be challenges in implementing border screening, as infected travellers with short trip lengths may not have RDT-detectable levels of parasite density upon entry to Zanzibar. In addition, treating 75% to 100% of infected travellers is likely not feasible without more drastic measures such as mass drug administration to travellers. Targeting interventions such as chemoprophylaxis or awareness campaigns towards travellers to or from high-risk areas within mainland Tanzania may be more feasible and cost-effective. Further research in this area is needed to better quantify what these effects may be.

In general, we see that combinations of interventions have a compounding effect on incidence. For example, improvements to RCD such as a combination of switching to RDA, following up all index cases promptly, and increasing the rate at which infected individuals seek treatment, can lead to large declines in malaria incidence on both islands (50% reduction on Pemba and 33% reduction on Unguja). We see that including neighbours leads to relatively small gains as the frequency of infections amongst neighbours was found to be very low in the RADZEC survey data, similar to the general population prevalence[11]. Including 100 neighbours in RCD would require a large amount of extra effort on the part of surveillance officers as many neighbouring houses would need to be visited and many more tests would need to be conducted. This result is in line with a previous modelling study that used an individual-based model for malaria to investigate the relationship between the search radius and the entomological inoculation rate (EIR)[14]. Reiker et al. (2019) found that at low EIR, increasing the search radius (i.e. the number of neighbours tested and treated) made no difference to the time to elimination[14].

The results shown here only consider stochastic uncertainty in the model. When uncertainty in the parameter values used is also included, the median final prevalence reached in each of the intervention scenarios remains the same but the confidence intervals widen, with substantial overlap. Nonetheless, the probability of reaching elimination does not change substantially. Details on how parameter uncertainty was included and the results from this analysis can be found in section S2.2 of the Supplementary Information.

RDA would likely confer some kind of a prophylactic effect in individuals given presumptive treatment, and may thus lead to a larger impact than that modelled here. On the other hand, since our model does not include acquired immunity, we assume that all malaria infections are equally likely to transmit malaria, regardless of parasite density. There is some evidence to suggest that individuals with lower parasitemia, who are more likely to show up as negative on an RDT, have lower gametocytemia and thus are less infective to mosquitoes

than RDT-positive individuals[15,16]. In this case, the impact of RCD may be underestimated by the model and the impact of a switch to RDA may be overestimated.

A previous study from Zambia found that the targeting ratio, the ratio of malaria prevalence in those tested and treated in RCD as compared to the general population, increases with decreasing prevalence, i.e. the clustering of infections increases as prevalence falls[17]. In this study, since we had no data on the impact of changing prevalence on the targeting ratio, we assumed a constant targeting ratio. In section S2.4 of the Supplementary Information, we compare the impact of a fixed targeting ratio to one that varies according to the function fitted in Chitnis et al. (2019). We find a minor improvement in the impact of RCD with a targeting ratio that increases with decreasing prevalence.

We define the term imported infection as relative to the patch of residence, where patch refers to either Pemba, Unguja or mainland Tanzania. Thus, only residents of Pemba or Unguja who are infected while away from their patch of residence are counted as imported infections on Zanzibar in the model. In terms of reporting, it is likely that a resident of mainland Tanzania who experiences malaria symptoms and seeks treatment while they are in Zanzibar would be classified and recorded as an imported infection in Zanzibar. Thus, we do not expect our model's estimates of imported infections to necessarily match with local records. Indeed, in the ZAMEP 2019–2020 Annual Report, it is estimated that 43% of cases are imported cases within the Malaria Case Notification database[18]. In comparison, we estimate that ~13% of new malaria infections amongst Zanzibari residents were acquired outside of Zanzibar in 2017–18. This discrepancy may arise due to a number of reasons, such as a change in travel patterns or malaria transmission rates from 2017 to 2020, or because cases may be acquired locally but still reported as imported if there is a history of travel, or because of a large number of mainland Tanzania residents seeking treatment for malaria while on Zanzibar. Such infected visitors would not be counted as imported cases within our model. However, transmission from such infected visitors is included as leading to introduced infections if they infect a local resident on the patch they are visiting, and so our estimates of introduced and indigenous infections match WHO definitions[3]. At any given time, the number of imported infections and the number of infected visitors on each of the three patches were estimated to be similar, so they contribute similarly to new infections (see Table S3 in the Supplementary Information). Therefore, roughly half of the introduced infections can be attributed to transmission from imported infections and half to transmission from infected visitors. In addition, as infected visitors contribute to the force of infection in the area that they are visiting, they can infect a susceptible traveller from the same area of residence as themselves. For example, two travellers from patch $k$, one susceptible and one infected, may travel together and transmission may occur between them when on patch $j$. In the model, the newly infected person would be counted as an imported case on patch $k$. This follows from the fact that transmission occurred via vectors on patch $j$, and imported cases are defined as cases arising from transmission away from the area of interest.

In these simulations, we assume that transmission restarts upon the incidence of a single indigenous infection. However, WHO allows for the presence of some indigenous infections after certification of elimination, as long as there are not more than three indigenous infections in one focus per year over 3 consecutive years[2]. As of yet, no country that has been certified malaria free has lost this status, suggesting that once elimination is reached, community transmission rarely restarts. A comparison of a transient and a cumulative probability of elimination is included in section S2.5 of the Supplementary Information.

This model assumes homogeneous mixing in each patch, with all individuals in a patch equally likely to become infected or to transmit an infection. However, heterogeneous biting rates would lead to a variation of the reproduction number within each patch[19,20]. Including

such heterogeneity is likely to make elimination even more difficult than our analysis suggests. However, heterogeneity in travel risk may make it easier to target travellers from high endemicity areas and allow for a larger impact on transmission with lower coverage.

In conclusion, the results of this study suggest that the largest group of infections on both major islands of Zanzibar are indigenous infections despite each infection typically leading to fewer than one new infection on both islands (i.e. the controlled reproduction number is estimated to be below 1 on both islands). The malaria burden on Zanzibar can be reduced substantially through a combination of interventions such as improvements to RCD and targeting treatment, chemoprophylaxis and bite avoidance measures towards travellers importing infections from mainland Tanzania. However, malaria elimination on Zanzibar will be difficult to achieve without a reduction in malaria prevalence on mainland Tanzania, highlighting the need for a coordinated effort within the United Republic of Tanzania to achieve elimination.

## Methods

We extend a stochastic metapopulation model described in ref. [10] to include separate compartments for imported, introduced and indigenous malaria infections. The model is parameterised to malaria prevalence and travel history data from the Reactive Case Detection: System Effectiveness and Cost (RADZEC) study conducted on Zanzibar and Malaria Atlas Project estimates of malaria prevalence for mainland Tanzania[11,21,22]. Data from a cross-sectional survey conducted during RCD, and extended to neighbours and a transect of households extending from the index household, inform the estimates of the population prevalence and increase in prevalence in index households and neighbouring households[11]. Results from a data audit conducted on the Malaria Case Notification register of Zanzibar inform estimates of the number of clinical cases typically reported at health facilities and the proportion of cases followed up[21]. The population prevalence at baseline for Pemba and Unguja is estimated by the prevalence of qPCR-detectable infections in neighbouring and transect households. The results from Micheweni, Pemba, from this dataset was compared to the PCR-detectable prevalence in a random sample in Micheweni in another study, and was found to be comparable[9–11]. Malaria Atlas Project estimates of malaria prevalence in 2–10-year-olds for the whole of Tanzania was used as the baseline prevalence on mainland Tanzania, as the RADZEC data on travel to mainland Tanzania suggested that residents of Zanzibar travel to many parts of mainland Tanzania, so the overall prevalence for Tanzania was taken in order to not assume travel to specifically high or low prevalence areas[11,22]. Further details of data collection can be found in Stuck et al. (2020) and van der Horst et al. (2020), and details of parameterisation can be found in Das et al. (2022)[10,11,21].

The model is based on a system of ordinary differential equations that include susceptible and infected humans in three patches, representing the islands of Pemba and Unguja, and mainland Tanzania. We include short-term human movement between the patches. Amongst infected humans, there are separate compartments for imported, introduced and indigenous infections on each patch. A schematic of the model is shown in Fig. 6.

### Movement model

If we first consider that there is just one patch and no human movement, but rather a constant rate of imported infections, the rate of change of imported infections can be described by:

$$\frac{dP}{dt} = \delta - \mu P, \tag{1}$$

where $P$ is the number of imported infections, $\delta$ is the rate of importation per unit time, and $\mu$ is the recovery rate.

**a**

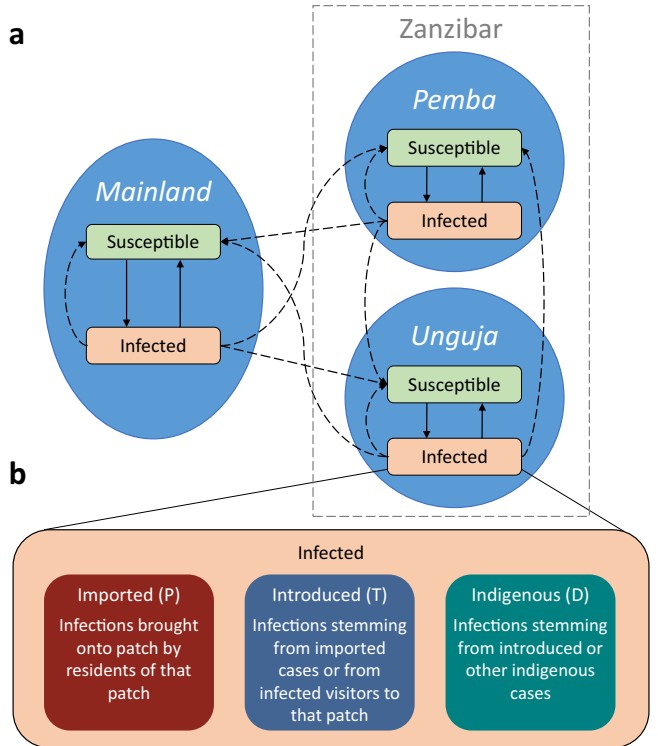

**b**

**Fig. 6 | Description of model patches and compartments, including sub-compartments for different categories of infections. a** A schematic diagram of the model with two disease states in each patch. Solid arrows represent transitions between disease states, and dashed arrows represent transmission. **b** A diagram of how the infected compartment is further divided into three sub-compartments comprising of imported, introduced and infected infections. Letters in brackets indicate state variable name in equations. These sub-compartments exist for all three patches.

If these imported infections then transmit the infection to other susceptible residents, the new infections would be classified as introduced infections according to WHO. The rate of change of introduced infections can be described by:

$$\frac{dT}{dt} = \beta P \frac{S}{N} - \mu T,\qquad(2)$$

where $T$ is the number of introduced infections, $\beta$ is the malaria transmission rate, $S$ is the number of susceptible residents, and $N$ is the total number of residents.

Further transmissions from these introduced infections lead to the second generation of infections from the original imported infections and are classified as indigenous infections by WHO. Similarly, further transmissions from indigenous infections lead to more indigenous infections. Thus, the rate of change of indigenous infections can be described by:

$$\frac{dD}{dt} = \beta(T+D)\frac{S}{N} - \mu D,\qquad(3)$$

where $D$ is the number of indigenous infections.

We then combine this framework for classifying infections into separate categories depending on where they are acquired and their position in the chain of transmission with a model for describing the movement of infection between patches[10]. Human mobility can be modelled with either an Eulerian perspective (where hosts explicitly move between patches) or a Lagrangian perspective (where hosts are fixed to their patch but can transmit infection between patches)[23].

In this model, we use a Lagrangian approach and label individuals by their patch, but allow them to contribute to transmission in other patches. The force of infection in any patch is therefore dependent on the malaria prevalence in all patches. This model is better suited to consider the short-term movement of people, which is expected to increasingly play a significant role in malaria persistence in low transmission areas[5–8,24]. As the median trip length in the Reactive Case Detection in Zanzibar: System Effectiveness and Cost (RADZEC) study was 6 days, we assume the majority of travel takes the form of short trips, and individuals retain the properties of their home patch[11].

We define a resident as someone who has lived in that patch for over 60 days, as the RADZEC travel data comes from questions regarding travel in the last 60 days. We define a visitor as someone who is temporarily visiting a patch other than their patch of residence. This is captured in the parameter $\theta_{ij}$, which gives the proportion of time the average resident of patch $j$ spends on patch $i$. Imported infections are defined as those where someone travelled away from their patch of residence, became infected with malaria while away, and then returned to their home patch infected. The rate of susceptible residents becoming imported infections is given by the proportion of the force of infection that they are exposed to when away from their home patch. Thus, the force of infection leading to imported infections in patch $k$, $\lambda_k^P$, is given by:

$$\lambda(t)_k^P = \sum_{i\neq k}^{n}\left(\beta_i\left(\frac{\sum_{j=1}^{n} N_j \theta_{ij} I(t)_j}{\sum_{j=1}^{n} N_j \theta_{ij}}\right)\theta_{ik}\right).\qquad(4)$$

We sum over all $i \neq k$ to get a total exposure away from home. In the context of Zanzibar and mainland Tanzania, the number of patches is set to 3, i.e. $n = 3$.

This is then combined with a recovery term that accounts for the natural clearance of infections and clearance due to reactive case detection to give:

$$\frac{dP_k}{dt} = \sum_{i\neq k}^{n}\left(\beta_i\left(\frac{\sum_{j=1}^{n} N_j \theta_{ij} I_j}{\sum_{j=1}^{n} N_j \theta_{ij}}\right)\theta_{ik}\right)S_k - (\mu + \varphi_k)P_k.\qquad(5)$$

Introduced infections in patch $k$ have either been infected from imported infections on patch $k$ (residents of $k$), or from visiting malaria infections who are residents of one of the other patches (who may be classified as an imported, introduced or indigenous infection on their patch of residence). Thus, the force of infection leading to introduced infections is given by the sum of the exposure of susceptible residents of $k$ to imported infections residing in patch $k$, and the exposure to infected individuals from other patches visiting patch $k$:

$$\lambda(t)_k^T = \beta_k\left(\frac{\theta_{kk}P(t)_k + \sum_{j\neq k}^{n} N_j \theta_{kj} I(t)_j}{\sum_{j=1}^{n} N_j \theta_{kj}}\right).\qquad(6)$$

The first term within the brackets is the contribution to the force of infection from imported infections amongst residents of patch $k$, and the second term is the contribution to the force of infection from all infected visitors who are visiting patch $k$ (hence, we sum over $j \neq k$).

Finally, when there is further transmission from introduced infections or indigenous infections that are residents of patch $k$ while they are in patch $k$, these lead to new indigenous infections. If they are not on patch $k$ during the time of transmission, they would lead to introduced infections in another patch if they infect a resident of that patch, or an imported infection if they infected another visitor to that patch. Thus, the force of infection term leading to indigenous infections is:

$$\lambda(t)_k^D = \beta_k\left(\frac{\theta_{kk}(T(t)_k + D(t)_k)}{\sum_{j=1}^{n} N_j \theta_{kj}}\right).\qquad(7)$$

**Table 3 | Descriptions of state variables, parameters, and derived parameters used in the model**

| State variable or parameter | Description and units |
|---|---|
| *State variables* | |
| $P_k$ | Number of imported infections in patch $k$. Humans. |
| $T_k$ | Number of introduced infections in patch $k$. Humans. |
| $D_k$ | Number of indigenous infections in patch $k$. Humans. |
| *Parameters* | |
| $N_k$ | Total number of people in patch $k$ (assumed to be constant). Humans. |
| $\beta_k$ | The effective malaria transmission rate from humans to other humans in patch $k$. Day$^{-1}$. |
| $\theta_{ij}$ | The proportion of time the average resident of patch $j$ spends in patch $i$. $\sum_i \theta_{ij} = 1 \, \forall j$. Dimensionless. |
| $\mu$ | Natural infection clearance rate. Day$^{-1}$. |
| $\tau_k^{(h)}$ | Ratio of malaria prevalence in the index household tested in RCD as compared to the general population in patch $k$. |
| $\tau_k^{(n)}$ | Ratio of malaria prevalence in neighbouring households tested in RCD as compared to the general population in patch $k$. |
| $\nu_k^{(h)}$ | Number of people tested in the index household during follow-up per index case in patch $k$. Dimensionless. |
| $\nu_k^{(n)}$ | Number of people tested in neighbouring households during follow-up per index case in patch $k$. Dimensionless. |
| $\rho$ | Rapid diagnostic test sensitivity. Dimensionless. |
| $\eta$ | The proportion of cases arriving at the health facility that are followed up. Dimensionless. |
| $\xi_k$ | The daily rate at which an infected individual seeks treatment in patch $k$. Day$^{-1}$. |
| *Derived parameters* | |
| $S_k$ | Number of susceptible humans in patch $k$, i.e. $S_k = N_k - P_k - T_k - D_k$. Humans. |
| $I_k$ | Proportion of humans who are infectious in patch $k$, i.e. $(P_k + T_k + D_k)/N_k$. Dimensionless. |
| $\varphi_k$ | Treatment rate due to RCD programme in patch $k$. Day$^{-1}$. |
| $\iota_k$ | Total number of cases arriving at a health facility in patch $k$. Humans per day. |

When the transmission terms for introduced and indigenous infections are also combined with recovery terms, the full sets of equations becomes:

$$\frac{dP_k}{dt} = \sum_{i \neq k}^{n} \left( \beta_i \left( \frac{\sum_{j=1}^{n} N_j \theta_{ij} I_j}{\sum_{j=1}^{n} N_j \theta_{ij}} \right) \theta_{ik} \right) S_k - (\mu + \varphi_k) P_k, \quad (8)$$

$$\frac{dT_k}{dt} = \beta_k \left( \frac{\theta_{kk} P_k + \sum_{j \neq k}^{n} N_j \theta_{kj} I_j}{\sum_{j=1}^{n} N_j \theta_{kj}} \right) \theta_{kk} S_k - (\mu + \varphi_k) T_k, \quad (9)$$

$$\frac{dD_k}{dt} = \beta_k \left( \frac{\theta_{kk}(T_k + D_k)}{\sum_{j=1}^{n} N_j \theta_{kj}} \right) \theta_{kk} S_k - (\mu + \varphi_k) D_k, \quad (10)$$

where $I_k = (P_k + T_k + D_k)/N_k$ for $k \in \{1, 2, 3\}$, i.e. the proportion of infected residents on each patch $k$, $n = 3$, and $\varphi_k$ represents the clearance rate due to RCD (this is described in more detail in Eq. (11)). State variable and parameter descriptions for Eqs. (8)–(10) can be found in Table 3.

The effective transmission rate, **β**, is estimated from the malaria prevalence, movement rates and RCD activities present on each of the three patches[10]. At equilibrium, the system of ordinary differential equations can be rearranged to a set of simultaneous equations, which can then be solved for **β** when **I**, **N**, **θ** and **φ** are known[10]. The transmission parameter, **β**, incorporates the baseline transmission potential, ongoing vector control activities, and ongoing passive surveillance.

**Reactive case detection**
RCD is a form of contact tracing where, due to the mosquito-borne nature of malaria, the focus is on geographically nearby contacts. Thus, nearby contacts of a known malaria case are followed up, tested for malaria, and treated if found to be positive. In Zanzibar, this involves following up index cases and testing and treating their household

members. The per capita rate of treatment due to RCD is:

$$\varphi_k(t) = \xi_k \eta \nu_k^{(h)} \tau_k^{(h)} I_k(t) \rho, \quad (11)$$

where $\xi_k$ is the rate at which infected individuals seek treatment at a health facility, $\eta$ is the proportion of index cases that are investigated at the index household level[21], $\nu_k^{(h)}$ is the size of the index household, $\tau_k^{(h)} I_k$ is the inflated prevalence amongst index household members, and $\rho$ is the rapid diagnostic test (RDT) sensitivity[11]. $\xi_k$ was derived from health facility data on the median number of malaria cases recorded per month per district on Pemba and Unguja, which was scaled by the number of districts on each island and 30 days in a month[21]. $\nu_k^{(h)}$ was estimated by calculating the mean index household size from RADZEC data[11]. $\tau_k^{(h)}$ was calculated by taking the mean number of infections found in an index household, dividing by the index household size, and taking the ratio of the prevalence in the index household to the malaria prevalence in the general population[11]. The baseline values for these parameters can be found in Table 4.

When neighbours are also included in RCD, the rate of treatment due to RCD is modified to the following:

$$\varphi_k(t) = \xi_k \eta (\nu_k^{(h)} \tau_k^{(h)} + \nu_k^{(n)} \tau_k^{(n)}) I_k(t) \rho, \quad (12)$$

where the superscripts $(h)$ refer to the index household and $(n)$ refers to the neighbouring households.

The daily number of malaria cases recorded in health facilities is:

$$\iota_k = \xi_k I_k N_k, \quad (13)$$

where $I_k N_k$ is the total number of infected people on patch $k$ and $\xi_k$ is the rate at which each infected person seeks treatment at a health facility and is diagnosed with malaria, as described earlier. We note that this rate is relatively low since many infections are likely to be asymptomatic and may never seek treatment in the course of the infection. The daily number of malaria cases recorded at a health facility is estimated from data on the median number of cases reported per district per month on Pemba and Unguja[21]. By assuming that this is

**Table 4 | Variable and parameter values at baseline and sources**

| Variable or parameter | Pemba | Unguja | Mainland | Source |
|---|---|---|---|---|
| $\iota_k^*$ | 1.36% | 1.18% | 7.79% | 11, 22 |
| $N_k$ | 406,848 | 896,721 | 43,625,354 | 26 |
| $\theta_{ij}$ | | Pemba Unguja Mainland | | 11 |
| | Pemba | 0.991   0.004   $5.7\times10^{-5}$ | | |
| | Unguja | 0.003   0.970   $5.3\times10^{-4}$ | | |
| | Mainland | 0.006   0.026   0.999 | | |
| $\mu$ | 0.005 day$^{-1}$ | 0.005 day$^{-1}$ | 0.005 day$^{-1}$ | 27, 28 |
| $\tau_k^{(h)}$ | 3.2 | 10.0 | N/A | 11 |
| $\tau_k^{(n)}$ | 0.7 | 1.3 | N/A | 11 |
| $\nu_k^{(h)}$ | 7.0 | 6.3 | N/A | 11 |
| $\nu_k^{(n)}$ | 20.4 | 18.8 | N/A | 11 |
| $\rho^*$ | 34% | 34% | N/A | 11 |
| $\eta^*$ | 35.3% | 35.3% | N/A | 21 |
| $\xi$ | $2.9\times10^{-4}$ day$^{-1}$ | $6.1\times10^{-4}$ day$^{-1}$ | N/A | 11, 21 |

The superscripts $(h)$ indicate the index household and $(n)$ indicates neighbouring households.

the value of $\iota_k^*$ at baseline and assuming equilibrium prevalence, we estimate that the treatment-seeking rate is:

$$\xi_k = \frac{\iota_k^*}{I_k^* N_k}, \qquad (14)$$

where an asterisk indicates the value of that parameter at baseline.

### Model simulations
Equations (8)–(10) were simulated using a binomial tau-leap adaptation of the Gillespie algorithm[25]. The initial conditions were set such that all infections were indigenous infections, and then the model was run for ten years to allow it to reach an equilibrium of imported, introduced and indigenous infections. After this, interventions were introduced and simulations were run for another 40 years. Simulations were repeated 500 times to account for stochastic variation. In all figures, interventions are introduced in year 0, which is calibrated to data from 2017–18. Note, in some figures, results are displayed for the first 20 years from the start of interventions. Simulations were run using Python version 3.6.6 and Numba version 0.39.0. Figures were plotted in R version 4.1.2, using ggplot2 version 3.3.5.

### Model with interventions
The baseline model was expanded to include the following interventions:
1. RCD at a range of levels of case follow-up. At baseline, 35% of malaria cases diagnosed at a health facility are followed up at the index household level within 3 days[21].
2. RCD with follow-up of neighbouring households. Currently, neighbours are not generally included in RCD in Zanzibar. We test the impact of including individuals in neighbouring households in testing and treatment upon investigation of the index case.
3. Switching from RCD to RDA. Currently, an RDT is used to diagnose malaria in those followed up by RCD. This RDT is estimated to have a sensitivity of 34% as compared to qPCR due to a high frequency of low parasite density infections[11]. Switching to RDA means that the RDT is no longer used during follow-up and all members of the household are given presumptive treatment.
4. RCD at a range of levels of treatment seeking. At baseline, the rate of seeking treatment is $2.9\times10^{-4}$ per day in Pemba and $6.1\times10^{-4}$ per day in Unguja. We test the impact of increasing the treatment-seeking rate of infected individuals. For example, this could be due to waning immunity and thus a higher proportion of

symptomatic infections in the population, broader screening measures in health facilities, or including pharmacies or drug stores in the case notification system.
5. Treatment and prevention of a proportion of infections brought on to Zanzibar by travelling humans (either residents or visitors). This could be through prevention measures such as chemoprophylaxis or bite avoidance measures for Zanzibari residents when visiting mainland Tanzania or treatment of mainland Tanzania residents on arrival at Zanzibar. In order to be concise, this intervention is referred to as treatment of travellers.
6. Reductions in the malaria transmission rate on each of the islands of Zanzibar, potentially through intensified vector control, i.e. $\beta_{\text{Zanzibar}}^{\text{intervention}} = \beta_{\text{Zanzibar}}(1 - r_{\text{Zanzibar}})$, where $r$ refers to the reduction in vectorial capacity, and the subscript Zanzibar refers to either Pemba or Unguja.
7. Reductions in the malaria transmission rate on mainland Tanzania potentially through intensified vector control, i.e. $\beta_{\text{Mainland}}^{\text{intervention}} = \beta_{\text{Mainland}}(1 - r_{\text{Mainland}})$.

Interventions 1 to 6 were applied simultaneously to the Pemba and Unguja patches. Interventions 1 to 4 are collectively referred to as RCD-related interventions. Baseline and intervention values simulated can be found in Table 2.

Details of how the interventions were included in the model are described in Section S1.1 of the Supplementary Information.

### Reporting summary
Further information on research design is available in the Nature Portfolio Reporting Summary linked to this article.

### Data availability
The data and code needed to run this model are available on GitHub and deposited in the Zenodo database under accession code https://doi.org/10.5281/zenodo.7782511. Publicly available data from the Malaria Atlas Project was also used to parameterise the model. This data can be found at https://data.malariaatlas.org. No data was specifically collected for this study.

### Code availability
The data and code needed to run this model are available on GitHub and deposited in the Zenodo database under accession code https://doi.org/10.5281/zenodo.7782511. Modelling, data analysis and plotting were conducted using Python version 3.6.6, numba version 0.39.0, R version 4.1.2 and ggplot2 version 3.3.5.

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

## Acknowledgements

We would like to thank Kim Lindblade, Lars Kamber, Aurélien Cavelan, Clara Champagne, Emma Fairbanks, Thiery Masserey and Pascal Grobecker for their helpful discussions and feedback on the manuscript. We would like to thank Faiza Abbas for her support of this project. Calculations were performed at the sciCORE (http://scicore.unibas.ch/) scientific computing center at University of Basel. A.M.D. and N.C. were supported by the Bill and Melinda Gates Foundation (INV025569). Funding for the RADZEC study was provided by the Swiss Tropical and Public Health Institute and the US President's Malaria Initiative via the US Agency for International Development/Tanzania under the terms of an inter-agency agreement with Centers for Disease Control and Prevention (CDC) and the US Agency for International Development/Tanzania through a cooperative agreement with the MEASURE Evaluation consortium, under the associate cooperative agreement No. AID-621-LA-14-00001 titled 'Measure Phase III-Strengthening the monitoring, evaluation and research capacity of the community health and social service programmes in the United Republic of Tanzania'. This grant supported the initial data collection and analysis conducted by M.W.H., J.O.Y., L.S. and B.S.F. The opinions expressed herein are those of the authors and do not necessarily reflect the views of the President's Malaria Initiative via the US Agency for International Development, or other employing organizations or sources of funding. BSF was additionally supported by a PhD scholarship from the Canton of Basel-Stadt, Switzerland.

## Author contributions

A.M.D. was responsible for study conceptualisation, developing the methods and data analysis, writing, editing and reviewing the manuscript. M.W.H. and J.O.Y. were responsible for study conceptualisation, and contributed to writing, reviewing and editing the manuscript. L.S., B.S.F., A.H.A. and A.A. were responsible for collecting the data used in this study, and contributed to writing, reviewing and editing the manuscript. N.C. was responsible for study conceptualisation, developing the methodology, supervision, and contributed to writing, reviewing and editing the manuscript.

## Competing interests

The authors declare no competing interests.
