## [Peer Review File · Nature Communications]

Modelling the impact of interventions on imported, introduced and indigenous malaria infections in Zanzibar, TanzaniaREVIEWER COMMENTS

Reviewer #1 (Remarks to the Author):

Review of the manuscript entitled "Modelling the impact of interventions on imported, introduced and indigenous malaria infections in Zanzibar, Tanzania" for Nature Communications.

Summary

The authors present the results of a modelling study where they calibrate a compartmental model to malaria prevalence data and mobility data from Zanzibar, Tanzania. The model explicitly accounts for imported, indigenous and introduced cases which is novel and allows the exploration of transmission reduction scenarios in relation to the WHO guidelines for elimination. They show that due to the high proportion of imported cases into Zanzibar, elimination efforts will need to be nation-wide.

The manuscript reads well and shows interesting findings. The authors show the importance of their work and make clear conclusions and suggestions for future elimination targets. It is great to see that the authors make the code available via github. I have some concerns about the ambiguity of the vector control intervention described (discussed below), and a few suggestions to improve the presentation of results.

Major revisions

1. The model for each patch has S and I compartments, but not recovered. I am not a malaria expert so this could be wrong if infection does not confer long-term immunity, but should there be an R compartment for people who have recovered? If not, then perhaps some mention of temporary immunity levels in the discussion would be useful.
2. Reductions in transmission through vector control are explored as an intervention. However, the model does not use vector compartments and therefore cannot model this mechanistically. Instead, reductions in human-to-human transmission are used to represent increased vector control. More explanation is needed on how the quantitative reductions in human-to-human transmission are to be achieved and through which vector control strategies (currently it is too vague). Additionally, vector control in Zanzibar is already high, as mentioned in the discussion on page 9, so what does 'increased vector control' mean going forwards? If it is not possible to increase and cannot be quantified, then I would suggest removing 'vector control' as an intervention explored in the model.
3. Error bars show the 95% range of the results in Fig 1 and Fig 2. I am unclear what this means. Reporting the 95% confidence intervals instead would be more statistically robust.

Minor revisions/ suggestions

1. Explicitly defining what is meant by indigenous, imported and introduced cases (as is shown in Fig 6) earlier in the introduction would help readers from non-malaria backgrounds.
2. The last part of the introduction would be more suitably placed in the methods section. For the introduction, a simpler overview of the interventions tested in the model would be better than writing out each combination.
3. A map of Zanzibar would be a useful visual aid, perhaps with some demographic details about the two islands, for example, differences in mobility between the mainland and the 2 islands, or current prevalence.
4. '0' and 'zero' are used interchangeably
5. The start of the results section shows estimates for the proportion of cases but does not say for which time point, I assume the estimates are for the current year at $t=0$, but which year is that? Similarly, in the data description of the methods section, the years that data are available for needs reporting, to contextualise the model results in time.
6. In Fig 2, a couple of individual runs are shown on the plots. I would suggest either showing all the runs as faint lines behind the median and range, or not show any individual runs.
7. 'Patches' are mentioned in the discussion which comes before the explanation of what patch means in the methods section. A brief explanation of what patch means (either Pemba, Ungunja or mainland) when the term is first used is necessary.

Reviewer #2 (Remarks to the Author):

This is an interesting analysis, but the authors do not make clear as to what is novel regarding their results in comparison with earlier work, both that of the authors (reference 10 is an unpublished version of a simpler model by the authors), and by other authors.

Major Comments:

1. The definitions of the term imported infection is rather confusing, and appears to be different in different parts of the manuscript: is an imported infection a resident of Zanzibar who becomes infected on the mainland and/or an infected resident of the mainland that visits Zanzibar? Is this the authors terminology, or has it been defined by the WHO? If so, please provide a reference.
2. I suggest the authors present results for 15 years, rather than at the equilibrium value, which is reached after 40 years. Assuming that conditions will stay constant for 40 years is unrealistic.
3. The authors present an uncertainty and sensitivity analysis of parameter uncertainty. Additionally, the authors should present an uncertainty and sensitivity analysis of the interventions that they are investigating. This would allow the authors to quantitatively investigate a range of values for each intervention (rather than a few scenarios) and obtain a quantitative comparison of equivalence between interventions, and combinations of interventions. These results should be included in the body of the manuscript, and replace the current results.
4. Why did the authors only conduct relatively few (500) stochastic simulations? Why not 10,000?
5. The Equations should be rearranged. Equation 5 should be discussed before equation 4a. Equation 6 should be discussed before equation 4b. Equation 7 should be discussed before equation 4c.

Minor Comments:

1. Figures where the probability is zero should not be shown.
2. Probabilities should be presented as fractions, and not as percentages.
3. The results for the counterfactual scenario should be removed.

Reviewer #3 (Remarks to the Author):

The manuscript entitled 'Modelling the impact of interventions on imported, introduced and indigenous malaria infections in Zanzibar, Tanzania' used a modelling approach to assess the impact of various control measures on malaria infections in Zanzibar, Tanzania. The manuscript was well written, the methodologies were sound, and the results were fascinating. The work will be of significant interest to modellers of malaria, other vector-borne disease systems and policymakers/stakeholders in malaria control in East Africa. I am thus happy to recommend the article for publication subject to revisions.

1. The authors referred to reference 10 within the main text several times, particularly within the Methods. At the time of this review, this article was yet to be published, so it is unclear what the details of the model are (namely parametrisation). Regardless of whether article 10 will be published, it is essential to outline the methods fully to readers in order for results to be reproduced. Furthermore, full transparency of the methods will allow readers to place the results appropriately within context.
2. Equation 5 implicitly implied that infected individuals from location k may visit location i , and contribute to the force of infection of importations into location k itself. This should be stated explicitly. This could also be discussed—is a case truly imported if the infector is from the same island? What if the infector and susceptible travelled to j from k together?
3. The biological description of θ was included within Table 1, however it was not described

within the text. Including this description would be useful.

4. The full set of equations (equation 4) was stated before the human movement model was derived. Section 4.1.1 should come before equation 4.
5. What do the error bars represent in Figure 3? Violin, or beeswarm, plots will indicate, or fully show, outcome uncertainty, respectively (give the model design).
6. Although stated in the methodology, how many simulations of the model are each of the Figures based on? This is required as part of the Reporting Summary document ('The exact sample size (n) for each experimental group/condition, given as a discrete number and unit of measurement').
7. It would be beneficial to have some confidence intervals on the lines on Figures 4 and 5.
8. In the introduction, the authors stated that they do not consider induced infections as the 'vast majority' of cases are imported, introduced or indigenous. It would be useful for the reader if a quantitative result could be attached to this—what percentage of cases roughly?
9. In the attached reporting summary form, the authors have confirmed that 'For Bayesian analysis, information on the choice of priors and Markov chain Monte Carlo settings' is included. However, there is no indication in the manuscript that the authors have performed Bayesian analysis.

Reviewer #4 (Remarks to the Author):

This paper extended a stochastic metapopulation model (available as a pre-print elsewhere) of malaria transmission and human movement between Zanzibar and mainland Tanzania to estimate the proportion of malaria cases which are indigenous, imported and introduced in Zanzibar and explore the impact of several interventions as well as reduction of transmission in mainland Tanzania on progress towards elimination.

The authors estimate that the majority of new cases in Zanzibar are indigenous (88% in Pemba, 56% in Unguja) despite high levels of importation, and also estimate that, especially on Unguja island, extended RCD coverage and treatment of travellers could reduce cases, but reduction of the malaria burden in mainland Tanzania would be required to achieve elimination in Zanzibar.

This work will be relevant to elimination and pre-elimination countries considering RDA/RCD strategies, as well as academics working in malaria elimination. It is very specific to the context of Zanzibar but could be adapted to other locations provided there are sufficient data to parameterise and validate the model. The paper seems relatively reproducible and the underlying model code is also available.

My main concern is in the model fitting and validation. Predictions are being made over a 40 year timespan, and the model used has not been peer reviewed and did not seem to be extensively validated. Whilst the model has been parameterised to data, further details of this, as well as validation on additional real or simulated data would be helpful. It also would be helpful if the model was tested using simulated data to show that it could correctly infer proportions of indigenous, local and imported cases. If further validation has been carried out, this should be stated in this paper.

Specific comments below

1. Would be useful to include table of definitions in introduction
2. The authors explain that the model was parameterized using the RADZEC study, has there been

any validation outside of this dataset, and were any aspects of the modelling framework tested using simulated data? It seems most parameters are coming from a single paper/study and parameter fitting approaches are not clearly described

3. Reference 10 is a key reference but there is no link to a doi
2. S2.4 in the supplement: 'targeting' has a typo

Response to reviewers

We thank the reviewers for taking the time to review our manuscript and providing us with valuable feedback. Please find below the original reviewer remarks in red and our responses in black.

Reviewer #1 (Remarks to the Author):

Review of the manuscript entitled "Modelling the impact of interventions on imported, introduced and indigenous malaria infections in Zanzibar, Tanzania" for Nature Communications.

Summary

The authors present the results of a modelling study where they calibrate a compartmental model to malaria prevalence data and mobility data from Zanzibar, Tanzania. The model explicitly accounts for imported, indigenous and introduced cases which is novel and allows the exploration of transmission reduction scenarios in relation to the WHO guidelines for elimination. They show that due to the high proportion of imported cases into Zanzibar, elimination efforts will need to be nation-wide. The manuscript reads well and shows interesting findings. The authors show the importance of their work and make clear conclusions and suggestions for future elimination targets. It is great to see that the authors make the code available via github. I have some concerns about the ambiguity of the vector control intervention described (discussed below), and a few suggestions to improve the presentation of results.

Major revisions

1. The model for each patch has S and I compartments, but not recovered. I am not a malaria expert so this could be wrong if infection does not confer long-term immunity, but should there be an R compartment for people who have recovered? If not, then perhaps some mention of temporary immunity levels in the discussion would be useful.

Infection with malaria does not convey sterile immunity and predominantly reduces parasite densities of successive infections. The SI model ignores this effect of acquired immunity and we now explicitly refer to this in the Discussion paragraph where we describe the implications on onwards transmission of malaria: *On the other hand, since our model does not include acquired immunity, we assume that all malaria infections are equally likely to transmit malaria, regardless of parasite density. There is some evidence to suggest that individuals with lower parasitemia, who are more likely to show up as negative on an RDT, have lower gametocytemia and thus are less infective to mosquitoes than RDT positive individuals (Slater 2015, Kobayashi 2019). In this case, the impact of RCD may be underestimated by the model and the impact of a switch to RDA may be overestimated.* Furthermore, as transmission decreases, the level of acquired immunity decreases and malaria dynamics become closer to SIS-type dynamics.

2. Reductions in transmission through vector control are explored as an intervention. However, the model does not use vector compartments and therefore cannot model this mechanistically. Instead, reductions in human-to-human transmission are used to represent increased vector control. More explanation is needed on how the quantitative reductions in human-to-human transmission are to be achieved and through which vector control strategies (currently it is too vague). Additionally, vector control in Zanzibar is already high, as mentioned in the discussion on page 9, so what does 'increased vector control' mean going forwards? If it is not possible to increase and cannot be quantified, then I would suggest removing 'vector control' as an intervention explored in the model.

We agree that we did not model vector control explicitly and have removed references to vector control from the Introduction and Results to focus on transmission reduction. In the introduction, we added the following sentence, “*We also considered the impact of further reductions in transmission rates, both on Zanzibar and on the mainland, although we did not explicitly model the interventions that would lead to the reductions.*”

We further replaced the discussion of vector control in the Discussion to start with the focus on transmission reduction and expand on how it would be achieved with, “*Simulated decreases in the transmission rates on both Zanzibar and mainland Tanzania led to the largest reduction in malaria incidence and the highest likelihood of achieving malaria elimination on Zanzibar. Given that insecticide-treated nets and indoor residual spraying are already widely deployed in Zanzibar, further decreases in transmission rates may be difficult, but could potentially be achieved through novel supplementary vector control interventions such as volatile pyrethroid spatial repellents, odour-baited traps, and attractive targeted sugar baits. Transmission reduction could also be achieved with reactive vector control, which has shown promise in a field study in Namibia, especially when used in combination with RDA, and could be considered for deployment in a setting like Zanzibar (Hsiang 2020).*”

In the methods, we clarified that reductions in transmission rates could ‘*potentially*’ be achieved through vector control.

3. Error bars show the 95% range of the results in Fig 1 and Fig 2. I am unclear what this means. Reporting the 95% confidence intervals instead would be more statistically robust.

We have replaced range with ‘prediction interval’ since these values represent the 95% (correspondingly also interquartile in Fig. 2) range of where we expect simulation results to lie.

Minor revisions/ suggestions

1. Explicitly defining what is meant by indigenous, imported and introduced cases (as is shown in Fig 6) earlier in the introduction would help readers from non-malaria backgrounds.

The introduction text has been re-arranged to address this point.

2. The last part of the introduction would be more suitably placed in the methods section. For the introduction, a simpler overview of the interventions tested in the model would be better than writing out each combination.

This section has now been removed from the introduction and the descriptions of the interventions in the introduction has been slightly expanded to the following: *We then use this model to examine the impact of combinations of interventions such as improvements to reactive case detection (RCD), increasing the number of clinical cases detected in health facilities, switching to reactive drug administration (RDA), treatment of imported infections, and reductions in transmission through vector control.*

3. A map of Zanzibar would be a useful visual aid, perhaps with some demographic details about the two islands, for example, differences in mobility between the mainland and the 2 islands, or current prevalence.

We have added a paragraph giving a brief overview of Zanzibar and added a map of the islands in what is now Figure 1.

4. '0' and 'zero' are used interchangeably

All instances of '0 indigenous cases' have been changed to 'zero indigenous cases'. We have left all instances of 'year 0' as is.

5. The start of the results section shows estimates for the proportion of cases but does not say for which time point, I assume the estimates are for the current year at $t=0$, but which year is that? Similarly, in the data description of the methods section, the years that data are available for needs reporting, to contextualise the model results in time.

It has now been made clearer in the text that the data and baseline values are for 2017—18.

6. In Fig 2, a couple of individual runs are shown on the plots. I would suggest either showing all the runs as faint lines behind the median and range, or not show any individual runs.

We agree with the reviewer that the previous figure put too much emphasis on the individual runs and we have therefore turned them into faint lines. However, showing all 500 runs would make the plots difficult to read so we have only selected a few sample runs. We prefer to include at least some individual runs to show the stochastic variation within each simulation.

7. 'Patches' are mentioned in the discussion which comes before the explanation of what patch means in the methods section. A brief explanation of what patch means (either Pemba, Unguja or mainland) when the term is first used is necessary.

Thank you for pointing this out. We have now amended the discussion to now include: *We define the term "imported infection" as relative to the patch of residence, where 'patch' refers to either Pemba, Unguja or mainland Tanzania.*

Reviewer #2 (Remarks to the Author):

This is an interesting analysis, but the authors do not make clear as to what is novel regarding their results in comparison with earlier work, both that of the authors (reference 10 is an unpublished version of a simpler model by the authors), and by other authors.

We have now clarified in the Introduction what has been done previous and what is new in this study: modelling introduced and indigenous cases had not been done previously so this type of model and the application of the WHO definition of elimination is novel:

However, all these studies defined elimination as zero malaria infections, irrespective of their classification, which is not realistic in areas with regular movement of people to and from neighbouring regions with ongoing endemic transmission. To our knowledge, no prior studies have modelled imported, introduced, and indigenous infections explicitly and examined the impact of

interventions on these three categories of infections; therefore no previous work has been able to model the probability of elimination as defined by the WHO.

In this study, we explicitly model imported, introduced and indigenous separately to model the feasibility of achieving three years with no indigenous cases with current and potential future interventions to achieve the WHO standard for malaria-free certification.

Major Comments:

1. The definitions of the term imported infection is rather confusing, and appears to be different in different parts of the manuscript: is an imported infection a resident of Zanzibar who becomes infected on the mainland and/or an infected resident of the mainland that visits Zanzibar? Is this the authors terminology, or has it been defined by the WHO? If so, please provide a reference.

The definitions of imported, introduced and indigenous infections have now been defined in a Table in the Introduction and the differences between WHO and our definitions further clarified in the following paragraphs:

The World Health Organization (WHO) defines malaria elimination as "the interruption of local transmission of a specified malaria parasite species in a defined geographical area as a result of deliberate activities" (WHO Malaria Elimination). WHO defines the interruption of local transmission as "the reduction to zero incidence of indigenous cases", where it classifies Plasmodium falciparum malaria cases into the following categories: imported, introduced, indigenous, and induced, as defined in Table 1. Certification of malaria-free status by WHO requires the country to show three years of zero indigenous cases (WHO Malaria Terminology).

We follow WHO terminology in defining a malaria case as anyone infected with P. falciparum parasites, including both symptomatic and asymptomatic infections. However, we assume that diagnosis of cases only occurs in the patch of residence so we classify cases relative to their patch of residence: therefore we define imported infections as infections acquired when away from the area of residence; introduced infections as infections stemming from an imported infection, or from an infected visitor visiting the area of residence of the introduced infection; and indigenous infections as infections stemming from introduced or other indigenous infections. Thus, our definition of imported cases differs slightly from the WHO definition, as infected visitors are not counted as imported cases in the model (they would be classified as either an imported, introduced or indigenous case in their area of residence depending on where they acquired the infection). Our definition of introduced and indigenous cases match the definitions used by WHO, although in our simulations we have full knowledge of the chains of transmission, which is not always known by elimination programmes when classifying cases.

2. I suggest the authors present results for 15 years, rather than at the equilibrium value, which is reached after 40 years. Assuming that conditions will stay constant for 40 years is unrealistic.

We agree that conditions are unlikely to remain constant for 40 years and have changed the figures to show results over 15 years.

3. The authors present an uncertainty and sensitivity analysis of parameter uncertainty. Additionally, the authors should present an uncertainty and sensitivity analysis of the interventions that they are investigating. This would allow the authors to quantitatively investigate a range of values for each intervention (rather than a few scenarios) and obtain a quantitative comparison of

equivalence between interventions, and combinations of interventions. These results should be included in the body of the manuscript, and replace the current results.

We appreciate the sentiment of the reviewer. Although our results may be considered scenario analysis, they do cover a range (albeit discrete) of intervention values and provide us with a qualitative comparison of the interventions in reducing incidence of cases and achieving elimination according to the WHO definition. Expanding this range or converting it to a continuous range will refine our results but is unlikely to change the main message or results of the manuscript.

We also agree that a quantitative comparison of equivalence between interventions would be beneficial; however this would require a more sophisticated model of malaria transmission that captures more of the intricacies of the dynamics of malaria and the interventions. This is unfortunately out of the scope of this manuscript which is to provide a methodology for analysing the feasibility of malaria elimination as defined by the WHO using current and potential future interventions, applied to the case of Zanzibar.

4. Why did the authors only conduct relatively few (500) stochastic simulations? Why not 10,000?

We chose to run 500 simulations because this was sufficient to provide relatively narrow prediction intervals for the annual incidence and provide relatively smooth curves for the probability of elimination. Increasing the number of simulations would make these curves even smoother but would have little impact on the results of this analysis, and especially not the qualitative implications of these results.

5. The Equations should be rearranged. Equation 5 should be discussed before equation 4a. Equation 6 should be discussed before equation 4b. Equation 7 should be discussed before equation 4c.

This has now been done.

Minor Comments:

1. Figures where the probability is zero should not be shown.

We respectfully disagree with the reviewer here because in all figures, there are at least some curves that are nonzero and subplots showing values of only zero clearly illustrate the point elimination is not feasible for these parameter ranges.

2. Probabilities should be presented as fractions, and not as percentages.

We fully agree and have now replaced the axes to show numbers between 0 and 1.

3. The results for the counterfactual scenario should be removed.

The only counterfactual scenario we refer to in the Discussion is the scenario of removing RCD. We

believe this is important to include because it highlights that although RCD is unlikely to lead to elimination, stopping it would lead to a substantial increase in incidence and it is therefore important to continue with the program.

Reviewer #3 (Remarks to the Author):

The manuscript entitled 'Modelling the impact of interventions on imported, introduced and indigenous malaria infections in Zanzibar, Tanzania' used a modelling approach to assess the impact of various control measures on malaria infections in Zanzibar, Tanzania. The manuscript was well written, the methodologies were sound, and the results were fascinating. The work will be of significant interest to modellers of malaria, other vector-borne disease systems and policymakers/stakeholders in malaria control in East Africa. I am thus happy to recommend the article for publication subject to revisions.

1. The authors referred to reference 10 within the main text several times, particularly within the Methods. At the time of this review, this article was yet to be published, so it is unclear what the details of the model are (namely parameterisation). Regardless of whether article 10 will be published, it is essential to outline the methods fully to readers in order for results to be reproduced. Furthermore, full transparency of the methods will allow readers to place the results appropriately within context.

Reference 10 is now published. We have also added further details on the parameterisation, including ' ξ_k was derived from health facility data on the median number of malaria cases recorded per month per district on Pemba and Unguja, which was scaled by the number of districts on each island and 30 days in a month (van der Horst, et al., 2020). η^h was estimated by calculating the mean index household size from RADZEC data (Stuck, et al., 2020). T^h was calculated by taking the mean number of infections found in an index household, dividing by the index household size, and taking the ratio of the prevalence in the index household to the malaria prevalence in the general population (Stuck, et al., 2020). The baseline values for these parameters can be found in Table 2.'

2. Equation 5 implicitly implied that infected individuals from location $\$k\$$ may visit location $\$i\$$, and contribute to the force of infection of importations into location $\$k\$$ itself. This should be stated explicitly. This could also be discussed—is a case truly imported if the infector is from the same island? What if the infector and susceptible travelled to $\$j\$$ from $\$k\$$ together?

This is correct: the WHO definition of an imported case (as now explicitly reproduced in Table 1) is only contingent on the location of where the infection was acquired and not who it was acquired from. We have also added the following statement: '*Additionally, as infected visitors contribute to the force of infection in the area that they are visiting, they can infect a susceptible traveller from the same area of residence as themselves. For example, two travellers from patch $\$k\$$, one susceptible and one infected, may travel together and transmission may occur between them when on patch $\$j\$$. In the model, the newly infected person would be counted as an imported case on patch $\$k\$$. This*

follows from the fact that transmission occurred on patch j , and imported cases are defined as cases arising from transmission away from the area of interest.'

3. The biological description of θ was included within Table 1, however it was not described within the text. Including this description would be useful.

The description has now been included in main body of paper, in the section titled 'Movement model'

4. The full set of equations (equation 4) was stated before the human movement model was derived. Section 4.1.1 should come before equation 4.

This section has now been rearranged to include the full set of equations after section 4.

5. What do the error bars represent in Figure 3? Violin, or beeswarm, plots will indicate, or fully show, outcome uncertainty, respectively (give the model design).

The error bars represent the 95% range in simulation results and the following text has been added to the caption of Figure 3 (now Figure 4): *The error bars represent the 95% range in the annual incidence.* We have also added violin plots to Figure S2, which includes both stochastic and parameter uncertainty. We have kept the bar charts in the figure in the main text, since the error bars in this case only show stochastic uncertainty and are relatively narrow, to emphasise the median results.

6. Although stated in the methodology, how many simulations of the model are each of the Figures based on? This is required as part of the Reporting Summary document ('The exact sample size (n) for each experimental group/condition, given as a discrete number and unit of measurement').

The results in the figures are based on 500 stochastic simulations for each set of intervention parameters. As this is stated in the methodology, this has now been changed in the Reporting Summary document.

7. It would be beneficial to have some confidence intervals on the lines on Figures 4 and 5.

We thank the reviewer for this suggestion and have included confidence intervals in these plots and added a subsection to the Methods where we explain the calculation of the confidence intervals.

8. In the introduction, the authors stated that they do not consider induced infections as the 'vast majority' of cases are imported, introduced or indigenous. It would be useful for the reader if a

quantitative result could be attached to this—what percentage of cases roughly?

Induced cases are less than 0.1% of all classified cases in Zanzibar since 2013. Unfortunately ZAMEP case classifications are not publicly available so we can only cite this as personal communication. We have added the following sentence: *We do not include induced cases because they are responsible for less than 0.1% of all classified cases in Zanzibar (personal communication, ZAMEP).*

9. In the attached reporting summary form, the authors have confirmed that 'For Bayesian analysis, information on the choice of priors and Markov chain Monte Carlo settings' is included. However, there is no indication in the manuscript that the authors have performed Bayesian analysis.

This box has now been unticked. Additionally, we have now added the following description to the Supplementary Information, to clarify how the parameter distributions for the uncertainty analysis were derived: *Simulations were run with a range of parameter values based on the uncertainty in the data, taking the posterior distribution when an uninformative prior is updated with the observed data.*

Reviewer #4 (Remarks to the Author):

This paper extended a stochastic metapopulation model (available as a pre-print elsewhere) of malaria transmission and human movement between Zanzibar and mainland Tanzania to estimate the proportion of malaria cases which are indigenous, imported and introduced in Zanzibar and explore the impact of several interventions as well as reduction of transmission in mainland Tanzania on progress towards elimination.

The authors estimate that the majority of new cases in Zanzibar are indigenous (88% in Pemba, 56% in Unguja) despite high levels of importation, and also estimate that, especially on Unguja island, extended RCD coverage and treatment of travellers could reduce cases, but reduction of the malaria burden in mainland Tanzania would be required to achieve elimination in Zanzibar.

This work will be relevant to elimination and pre-elimination countries considering RDA/RCD strategies, as well as academics working in malaria elimination. It is very specific to the context of Zanzibar but could be adapted to other locations provided there are sufficient data to parameterise and validate the model. The paper seems relatively reproducible and the underlying model code is also available.

My main concern is in the model fitting and validation. Predictions are being made over a 40 year timespan, and the model used has not been peer reviewed and did not seem to be extensively validated. Whilst the model has been parameterised to data, further details of this, as well as validation on additional real or simulated data would be helpful. It also would be helpful if the model was tested using simulated data to show that it could correctly infer proportions of indigenous, local and imported cases. If further validation has been carried out, this should be stated in this paper.

We fully agree with the reviewer that validation is important. To address the first few points, we have now reduced the simulation time span to 15 years (also in response to a comment by Reviewer 2) and added details of the parameterisation (as described above in the response to Reviewer 3). The previous model has also passed peer review and is now published with an updated citation. The equations are fully described in the paper and the model code is openly available.

Again, we fully agree with the principle of validation but unfortunately the data needed to properly validate the model is unavailable since it depends on routinely collected data which is rarely publicly available. Furthermore, this would require consistent classification of cases. There are also no similar models that would allow us to validate the model with simulated data.

However, validation of the impact of changing intervention strategies on clinical incidence should be possible and such data would likely be available after Zanzibar switches to RDA, as currently planned. We hope to be able to conduct this validation after the switch.

Specific comments below

1. Would be useful to include table of definitions in introduction

We have included the WHO definitions of imported, introduced and indigenous cases in a table in the Introduction.

2. The authors explain that the model was parameterized using the RADZEC study, has there been any validation outside of this dataset, and were any aspects of the modelling framework tested using simulated data? It seems most parameters are coming from a single paper/study and parameter fitting approaches are not clearly described

Details of parameterisation have now been described in more detail (see response to first comment from Reviewer 3). We have further responded on validation in the main comment above.

3. Reference 10 is a key reference but there is no link to a doi

The reference has now been updated to the paper that has now been published in *Epidemics*.

4. S2.4 in the supplement: 'targeting' has a typo

All instances of 'targetting' in the supplementary information have now been changed to 'targeting'.

Bibliography

Das Aatreyee M. [et al.] The impact of reactive case detection on malaria transmission in Zanzibar in the presence of human mobility. *Epidemics*. - 2022. - Vol. 41. - p. 100639.

Stuck Logan [et al.] Malaria infection prevalence and sensitivity of reactive case detection in Zanzibar // *International Journal of Infectious Diseases*. - 2020. - Vol. 97. - pp. 337-346.

van der Horst Tina [et al.] Operational coverage and timeliness of reactive case detection for malaria elimination in Zanzibar, Tanzania. *The American Journal of Tropical Medicine and Hygiene*. - 2020. - 2 : Vol. 102.

World Health Organization WHO malaria terminology, 2021 update. World Health Organization, 2021.

REVIEWERS' COMMENTS

Reviewer #1 (Remarks to the Author):

Thank you for addressing my comments and for demonstrating where changes have been made. I am happy with the response and with the updated manuscript.

Reviewer #2 (Remarks to the Author):

All my comments/concerns from my previous review have been addressed.

Reviewer #3 (Remarks to the Author):

I would like to thank the authors for their careful and detailed response to the previous set of comments.

All of my concerns have been addressed. The manuscript was written effectively, the methodologies were robust, and the results were intriguing. This research will be of great value to malaria modelers, as well as other systems involving vector-borne diseases and stakeholders/policymakers involved in malaria control in East Africa.

Reviewer #4 (Remarks to the Author):

The authors have addressed my comments and concerns